# Miniature Autonomous Robot Based on Legged In-Plane Piezoelectric Resonators with Onboard Power and Control

**DOI:** 10.3390/mi13111815

**Published:** 2022-10-24

**Authors:** David Robles-Cuenca, Mario Rodolfo Ramírez-Palma, Víctor Ruiz-Díez, Jorge Hernando-García, José Luis Sánchez-Rojas

**Affiliations:** Microsystems, Actuators and Sensors Group, Universidad de Castilla-La Mancha, E-13071 Ciudad Real, Spain

**Keywords:** miniature robot, autonomy, piezoelectric, extensional mode, leg

## Abstract

This work reports the design, fabrication, and characterization of a centimetre-scale autonomous robot with locomotion based on in-plane piezoelectric resonators and 3D-printed inclined legs. The robot consists of a pair of cooperative piezoelectric motors, an electronic power circuit and a battery-powered microcontroller. The piezoelectric motors feature a lead zirconate titanate (PZT) plate of dimensions 20 mm × 3 mm × 0.2 mm vibrating on its first extensional resonant mode at around 70 kHz. A particular position of 3D-printed inclined legs allowed the conversion of the in-plane movement into an effective forward thrust. To enable arbitrary trajectories of the robot on a surface, two parallel piezoelectric plate motors were arranged in a differential drive scheme. The signals to excite these plates were generated by the microcontroller and adapted by a supplementary electronic circuit to increase the effective voltage supplied by the onboard battery. The fully assembled robot had a size of 27 mm × 15 mm and a weight of 7 g and reached a linear speed of approximately 15 mm/s and a rotational speed of up to 50 deg./s. Finally, the autonomous robot demonstrated the ability to follow pre-programmed paths.

## 1. Introduction

Autonomy is of paramount importance for the development of useful small mobile robots, projected to accomplish the tasks of standard-sized robots but in constrained volumes. As already reported [1], this field is full of challenges, starting from efficient methods of actuation for 2D movement to the integration of power supply and control interfaces on the robot-supporting platform.

Relating to actuation, the use of legs of different kinds is a common approach to achieve locomotion at the millimeter (mm) scale. Leaving aside sophisticated approaches that rely on legs with several degrees of freedom [2,3], a simpler approach is to consider the legs as passive elements that follow the movement of the robot body. Such movement might be a vibration induced by a rotatory motor [4], a standing wave, i.e., a vibration at one of the resonance frequencies of the system [5,6,7], or even a traveling wave by the mixing of standing waves [8]. The previous references were based on straight legs. Nevertheless, the literature also reported the use of inclined legs [9,10]. Recently, a so-called bristle-bot with mm size was reported with a piezoelectric patch to induce vertical actuation [11]. It is also worth mentioning that the use of bristles in combination with a resonant mode of vibration of the robot platform was reported before for applications such as inside tube inspection [12]. All the previous references relied on an actuation force perpendicular to the plane of the robot, i.e., in the vertical direction, intended to modulate friction through the variation of the normal force reaction during the cyclic movement of the legs, resulting in net locomotion.

Part of the actuation strategy is the challenge of steering the robot to obtain 2D movement. The development of various turning techniques has been studied, such as the individual control of the legs [13,14], tuning the signal frequency applied to the device [15,16] or a differential drive approach utilizing independent control of each side of the robot [3,17,18].

Autonomous robots require both onboard power and onboard control. Regarding onboard power, the classical approach is the use of batteries together with the required electronics to efficiently actuate the locomotion mechanism [19]. An alternative to batteries is the use of wireless tethers through fields, such as optical fields [20,21], magnetism [22,23] or vibration fields [24]. Regarding control, the combination of sensors with a microprocessor is a common strategy for closed-loop control. The robots named Alice [25] and Colias [26] were successful examples of wheel-based miniature autonomous robots with built-in power and control. Kilobot [27], DEAnsect [28] and HAMR-F [29] also demonstrated mm-scale autonomy, but with legged locomotion. 

Here, we report on a robot whose locomotion is based on the combination of an in-plane displacement of a plate resonator and inclined legs. No out-of-plane displacement was required; furthermore, the design here proposed minimized the presence of any out-of-plane contributions. The in-plane displacement corresponded to the first extensional resonant mode of a piezoelectric rectangular lead zirconate titanate (PZT) plate of mm-sized dimensions. The use of resonance increased the thrust applied to the legs. These were fabricated using flexible 3D-printed resins and attached at the edges of the piezoelectric plate. The inclination of the legs resulted in an asymmetry of the normal reaction force at the tip of the leg in contact with the ground, which translated into variations of the friction during the period of oscillation of the leg, leading to a net motion. To attain 2D locomotion, a common differential drive strategy was implemented using two parallel piezoelectric plates joined at their centers. This actuation mechanism could effectively carry the different loads associated with an autonomous robot: a battery and the electronics for driving the piezoelectric plates, plus a microcontroller board for open-loop control. A fabricated miniature 7 g robot reached a linear speed of approximately 15 mm/s and a circular speed of up to 50 deg./s and attained the capability to follow programmed paths.

## 2. Materials and Methods

### 2.1. Device Design

The fully assembled robot can be seen in Figure 1. The locomotion system was based on two parallel rectangular lead zirconate titanate (PZT) plates joined in an H-shaped configuration, with inclined legs at the edges of each plate. The electronic block consisted of a battery, a microcontroller board and a piezo drive circuit between the microcontroller and the PZT actuators.

The core of the motion mechanism relied on the first extensional mode of the rectangular PZT plates in combination with inclined legs (Figure 2a). The schematic shows a configuration with legs beneath the plate intended for walking and an identical geometry above the plate to realize a symmetric structure with respect to the plane of the plate. The purpose of this symmetry will be explained below. The dimensions of the plates were 20 mm long, 5 mm wide and 20 µm thick. Electrodes that covered the whole surface of the PZT plate ensured an optimized displacement [30]. Next, approximate analytical equations for the frequency and the displacement of the first extensional mode are presented. A free-free boundary condition was assumed, although the effect of the legs’ contact with the floor might affect this condition. Furthermore, the mechanical contribution of the electrodes was neglected, and both were considered short-circuited. Despite these assumptions, the following equations were useful for the development of the robot. Equation 1 describes the frequency f of the first extensional mode of a free-free plate [31]:(1)$$f=\frac{1}{2\cdot L_{s}}\sqrt{\frac{E}{\rho }}$$where $\;L_{s}$ is the length of the plate, and E and ρ are Young’s modulus and the density of the material, respectively. According to this equation, the frequency of the first extensional mode of a PZT plate with the previous dimensions is 70.4 kHz. This value was a point of reference for the experimental identification of the extensional mode. The in-plane displacement of the structure along the length of the plate can be estimated by Equation (2) [31]:(2)$$u=l\cdot \cos\left(\frac{\pi \cdot x}{L_{s}}\right)$$
where l is the maximum displacement, and x is the position along the structure. Figure 2b shows this displacement. No shift takes place at the centre, with maximum displacement at the edges, which suggests that the best location for the link between the two plates is in the middle of these.

Regarding legs, these were located only at the edges of the PZT plates supporting the structure to take advantage of the maximum displacement. 3D printed legs were fabricated using a flexible photopolymer with a density equal to 1.110 kg/m^3^ and a Young´s modulus equal to 1.750 MPa. Their dimensions were 0.5 mm long and 200 μm in diameter, with an angle of inclination of 60° with respect to the horizontal plane. A parallelepiped with the width of the plate, 0.5 mm thick in the longitudinal direction and 0.6 mm long normal to the plate, served as a support for the legs. Sets of 2, 3, 4 and 8 legs were considered along the width of the PZT plate without appreciable differences in performance during experimental tests. Therefore, 8 legs were used to better support the weight of the robot. Straight (vertical) legs were also tested and did not cause any motion, as expected.

To understand the proposed motion mechanism, simulations were carried out with the help of finite element analysis software [32]. Figure 3a shows the in-plane and the out-of-plane displacements along the top view of the PZT plate for the first extensional mode, with legs only at the bottom of the plate. Due to the asymmetry associated with the distribution of the legs, a clear out-of-plane component can be observed, which affects the trajectory of the legs and hence the performance of the robot. A symmetric arrangement of the legs with respect to the PZT plate was implemented to avoid this effect, as can be seen in Figure 3b, where a negligible out-of-plane displacement was obtained.

Figure 4 shows a side-view capture of the simulated total displacement of the plate with symmetric legs for the first extensional mode of the system. The colour scale represents in-plane displacement. When the plate goes leftward, the tip of the legs moves towards the plate, while the rightward movement of the plate pushes the legs away from the plate. The leg deformation towards the plate translates into closer proximity of the plate to the floor, increasing the normal reaction force. On the contrary, the deformation away from the plate elevates this and decreases the normal reaction force. Therefore, there is an asymmetry in the normal reaction force and hence in the friction force. This mechanism has been studied in [11] with actuation in the vertical direction, whereas in our case, the actuator is purely in-plane. In the figure under study, for the orientation of the legs shown, their movement to the left direction produces a higher friction force than to the right, and as a result, there is a net motion towards the right.

Next, the procedure to achieve 2D locomotion is described. Two PZT plates were joined together at their center in an H-shaped configuration by a 3D-printed beam designed to minimize coupling between both plates, which worked as independent resonators. To facilitate rotation, the separation of the plates reached the full width of the electronic board. Figure 5 shows the three possible ways of motion: forward motion with both plates actuated and either clockwise or counterclockwise rotation, depending on the plate to be actuated.

To allow the robot to follow a prescribed path, the three previous ways of motion should be adequately combined. The control signals were generated with a microcontroller board. This selection ensured the possibility of programming the proper routine for the PZT plate actuation, enabling the autonomous walking of the robot.

A common approach for the actuation of piezoelectric actuators on autonomous configurations consists of a battery, a boost stage to increase the battery voltage to the required level, and a driving stage that converts the output of the power stage into a time-varying signal to be applied to the actuator. The approach presented here did not require the boost converter, and the time-varying signal was directly obtained from the microcontroller pulse width modulation (PWM) output. Due to its low voltage amplitude, the 3.3 V square wave provided by the microcontroller board was not sufficient for the efficient actuation of the robot. Therefore, an interface amplification circuit was required between the microcontroller PWM output and the actuator terminal inputs. Figure 6a shows the schematic of this circuit, an idea proposed in reference [33]. The circuit requires an external inductance in parallel with the intrinsic capacitance of the piezoelectric actuator. The value of the inductance was chosen so that the resonance frequency of this LC circuit was twice the frequency of the resonance mode under interest. For the PZT plate, according to the manufacturer’s datasheet, a theoretical capacity of 4.62 nF was calculated. Following the mentioned criteria, a commercially available 470 μH coil was chosen. For the rest of the components, a PMEG6020ELRX diode, a BSR14 transistor and a resistor of 1 kΩ were used. Figure 6b shows the signals measured at the input (PWM from the microcontroller) and output, applied to the PZT plate.

Finally, with the purpose of controlling the energy transferred to the robot, the signal provided by the microcontroller was a burst-type signal (Figure 7). The first part of the signal was a 3.3 V square wave, with the frequency of the first extensional mode and duration T_ON_, and the second part was 0 V during T_OFF_. By varying either T_ON_, i.e., the number of cycles, or the period of the signal T_b_=T_ON_ + T_OFF_, the performance of the robot could be modified.

### 2.2. Robot Fabrication

The procedure for manufacturing the robot is explained below. The 20 mm long, 3 mm wide PZT plates (PIC 255 from PI Ceramic GmbH, Lederhose, Germany [34]) were manually cut from a larger PZT sheet 0.2 mm thick. A supporting structure (Figure 8) was 3D-printed with a stereolithography B9 Core printer (B9 Creations, Rapid City, SD, USA). The material for printing was black resin (B9 Creations, Rapid City, SD, USA), whose properties, once cured, were a density equal to 1.110 kg/m^3^ and a Young’s modulus equal to 1.750 MPa. The supporting structure had two purposes. First, it aimed to join together the two PZT plates at their mid-length, as explained in Figure 5. Second, it aimed to accommodate the rest of the robot hardware, i.e., the microcontroller board, the printed circuit board (PCB) with the amplification circuit, and the battery. The PZT plates were separated by 9 mm to keep them within the width of the board above and ensure a reliable rotation of the robot. The legs (Figure 8c) were also 3D-printed with the same printer and material mentioned above. A cyanoacrylate adhesive (Loctite, Düsseldorf, Germany [35]) was used to join the PZT plates to both the legs and the supporting structure.

A 3.6 V and 90 mAh Li-ion battery was used for the DC supply. This battery powered a Trinket microcontroller board (3V Trinket from Adafruit, NYC, NY, USA). The size of the board was 15 mm wide and 26.5 mm long, being the largest component of the total system. On top of this board, an auxiliary PCB was placed, featuring the amplification circuit of Figure 6. The electrical connection between the PZT plates and the PCB was attained with insulated copper wires 100 µm in diameter. Table 1 shows the mass of the parts of the robot and the total mass of the system, which was approximately 7 g.

### 2.3. Device Characterization

The electrical impedance of the PZT plates was measured with a 4294A Agilent impedance analyzer (Agilent Technologies, Santa Clara, CA, USA). A MEMSMap 510 system (Optonor, AS, Trondheim, Norway) allowed us to optically identify the mode of interest and determine the out-of-plane and the in-plane displacements. The maximum field of view of the equipment was 6 mm, so multiple measurements had to be taken and merged manually to cover the entire plates.

The kinetic performance of the robot was recorded using a high-speed and high-resolution camera (Logitech StreamCam, Lausanne, Switzerland). The videos were then processed by a Matlab (version R2021b, accessed on 27 July 2022) application (DLTdv digitizing tool) that tracked a pattern of 5 black points on a white piece of paper, attached to the robot. The information obtained from these points allowed us to calculate both the speed and the orientation of the robot. A Wycko Veeco optical profiling system was used to measure the roughness of the different surfaces on which the locomotion of the robot was tested.

## 3. Results

### 3.1. Electric Characterization

Figure 9 shows the measured electrical conductance of a PZT plate for three different cases: no leg, legs at the bottom and legs at both the top and the bottom. As expected, the added mass of the legs resulted in a shift to a lower frequency of the measured peaks.

The first point to notice is the appearance of a double peak when the legs were only attached to the bottom. Finite element analysis identified two neighbour modes for this situation, corresponding to the extensional mode previously mentioned in the design section (Figure 3a) with a significant out-of-plane contribution and different relative phases. Equally interesting is the measurement of only one conductance peak when the legs are arranged symmetrically with respect to the PZT plate. Figure 10 shows the experimental confirmation of the in-plane nature of this peak with the help of the Optonor optical system. In particular, Figure 10b displays a rather uniform colour without the characteristic lobes of the out-of-plane bending, in agreement with the simulated results (Figure 3b). It is important to mention that the apparent in-plane displacement measured at the legs was null, but this is an artifact as the 3D-printed support of the legs was out of focus. The legs’ movement was actually synchronized with the plate movement.

The electrical measurements were fitted to a Butterworth–Van Dyke equivalent circuit model, including the series resistance associated with the electrical connections [36]. Table 2 shows the values obtained from the fit for both the peak conductance and the quality factor, confirming the absence of deleterious effects in the symmetric-legged approach.

### 3.2. Kinectic Characterization

#### 3.2.1. Speed

Here, the results for the fully assembled robot are presented. For all the measurements, a frequency sweep around the measured conductance peak was performed to ensure that the selected actuation frequency provided the best performance. First, the locomotion with both motors actuated to describe a straight trajectory was studied. The burst-type signal mentioned before was used (Figure 7). The use of the same signal for both plates resulted in a curved path due to manufacturing tolerances and imperfections during the fabrication process. Therefore, while keeping the time T_b_ of the signal, the number of cycles, i.e., the time T_ON_, was varied for each plate. Figure 11a shows the path followed by the robot as a function of the ratio of the number of cycles applied to each plate. The robot started at point (0,0), and the ideal straight motion corresponded to the dotted line in the figure. According to the results, the ratio that best approached the straight line was 1:1.3. Once the straight motion was ensured, next, the linear speed was studied by varying the interval T_b_ of the signal while keeping the previous T_ON_ ratio (100 ms:130 ms) between plates. Figure 11b shows the results, with a top speed close to 15 mm/s. Appendix A, included in the Appendix A, shows the straight motion.

Next, the ability of the robot to rotate was tested following the procedure sketched in Figure 5. A burst signal of T_b_ equal to 1000 ms was used, and the T_ON_:T_b_ ratio was modified. Figure 12a,b show the results. It can be noticed that the counterclockwise speed is lower than the clockwise speed. This might be attributed to an uneven distribution of the mass of the robot, as well as to the nonuniform contact of the legs with the floor. Appendix A show the clockwise and counterclockwise rotation, respectively.

#### 3.2.2. Pulling Mass and Surface Roughness

The capability of the robot to pull masses was also studied. Different masses were tied to the robot and suspended by a pulley so that the robot pulled them as it moved forward. A drawing of the setup can be seen in the inset of Figure 13. Figure 13 shows the measured speed as a function of the pulled mass with a T_b_ = 250 ms and a ratio of T_ON_ between plates at 100 ms:130 ms. The robot stopped for masses greater than 300 mg (a blocking force of around 3 mN).

All previous speed measurements were made on a glass surface. In order to check the influence of the surface roughness on the speed, different materials were tested: glass, copper, aluminium, polyimide, 3D printed resin (black resin mentioned above), and paper; and the measured arithmetic average roughness was 5 nm, 346 nm, 762 nm, 66 nm, 594 nm, and 2.6 µm, respectively. The performance of the robot on metallic surfaces was comparable to the performance on glass. However, the robot no longer moved on polymer-based surfaces such as polyimide and 3D-printed resin. The same happened for paper. Further studies are in progress to analyze the interaction between the leg material and the contact surface material.

#### 3.2.3. Path Manipulation

The demonstration of the path manipulation of the robot was also accomplished. The path tracked was similar to a Z-shape so that the rotation of the robot in both directions could be checked. The proper actuation commands to the plates, with the help of the microcontroller board, navigated the robot through the path. No closed-loop control was implemented. Figure 14 shows the track followed by the robot, obtained with the previously mentioned software DLTdv, with snapshots at different instants. Appendix A shows the complete sequence.

## 4. Conclusions

This paper presents an autonomous miniature robot featuring a locomotion system with two piezoelectric plates vibrating in their first extensional mode and with attached inclined legs. This approach relies on the relative motion of the legs with respect to the horizontal plate, modulating friction and converting the cyclic oscillations into a net thrust, an innovation compared to locomotion systems based on vertical vibrations. The autonomy was implemented by a battery for the power supply and a microcontroller board to generate the square signals tuned to actuate the first extensional mode. To reach enough amplitude in the driving signal, a compact non-linear amplification circuit was used between the PWM outputs and the PZT plates.

The fully-assembled robot has a size of 27 mm × 15 mm and a weight of 7 g. Linear and angular speeds of nearly 15 mm/s and up to 50 degrees/s, respectively, were measured employing burst signals with appropriate timing and duration. Thus, the ability of the robot to follow a pre-programmed trajectory with precision was demonstrated.

## Figures and Tables

**Figure 1 micromachines-13-01815-f001:**
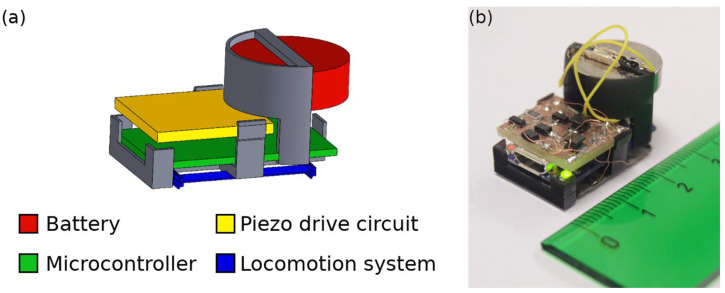
(**a**) Drawing with the parts of the robot. (**b**) Photograph of the fully-assembled robot. Ruler marked in cm.

**Figure 2 micromachines-13-01815-f002:**
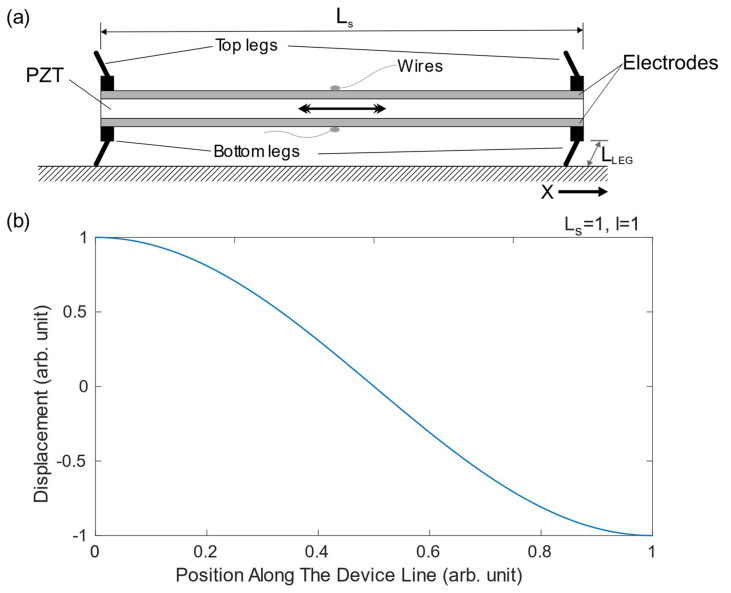
(**a**) Schematic of the locomotion system under study. (**b**) Normalized displacement in the X-axis for the first extensional mode.

**Figure 3 micromachines-13-01815-f003:**
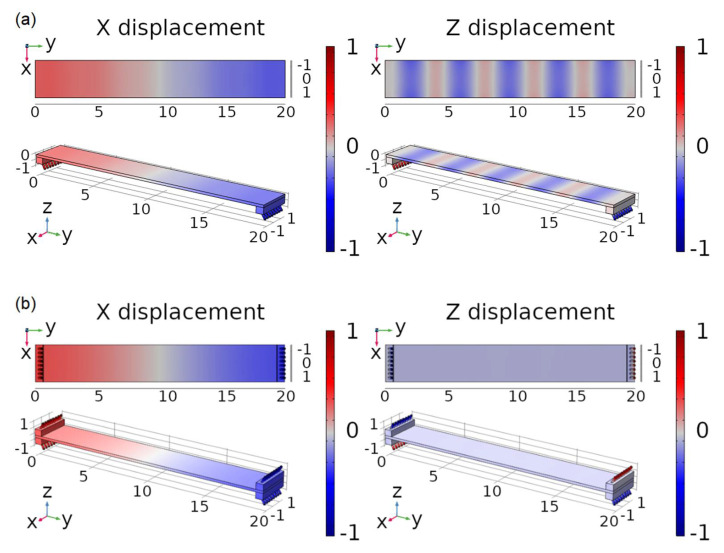
Top view and 3D view of the in-plane and out-of-plane normalized displacements of the first extensional mode of the PZT plate with (**a**) legs attached at the bottom and (**b**) legs attached at both the top and the bottom of the plate.

**Figure 4 micromachines-13-01815-f004:**
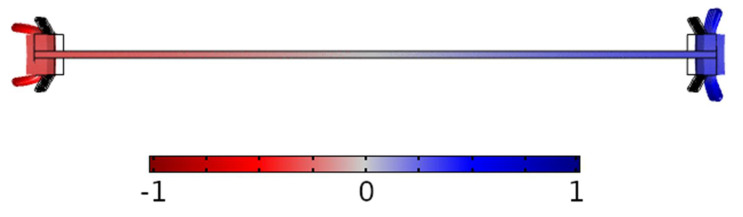
Side view capture of the normalized displacement of the first extensional mode of the PZT plate together with the symmetric legs configuration.

**Figure 5 micromachines-13-01815-f005:**
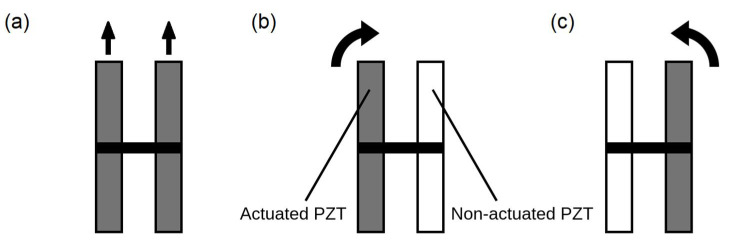
Top view of the H-shaped locomotion system with identification of the three possible ways of motion: (**a**) forward motion with both plates actuated, (**b**) clockwise rotation with left plate actuated, (**c**) counterclockwise rotation with right plate actuated.

**Figure 6 micromachines-13-01815-f006:**
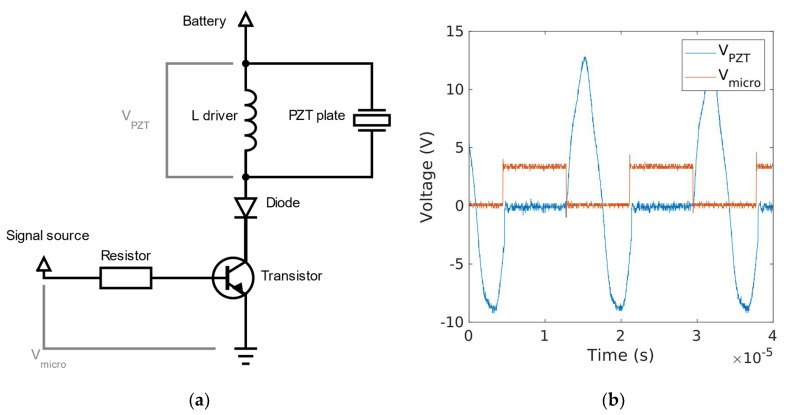
(**a**) Inductor-based interface circuit. (**b**) Measured PWM signal from microcontroller (orange) and voltage between PZT plate terminals (blue).

**Figure 7 micromachines-13-01815-f007:**
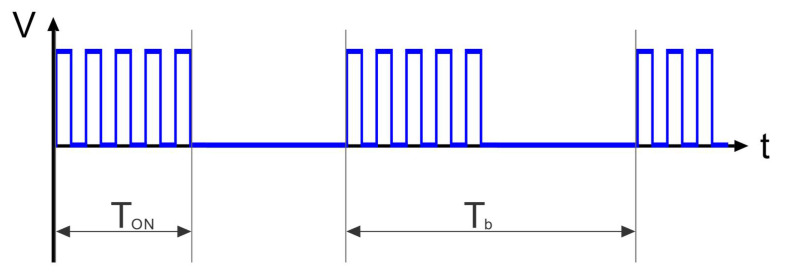
Actuation burst-type signal applied to the PZT plates.

**Figure 8 micromachines-13-01815-f008:**
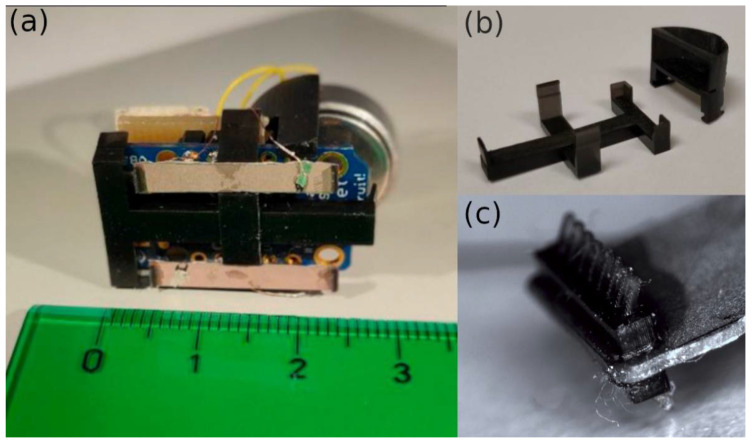
(**a**) Bottom view of the fully assembled robot. (**b**) 3D-printed supporting structure. (**c**) Detailed view of the legs on one of the edges of the PZT plate.

**Figure 9 micromachines-13-01815-f009:**
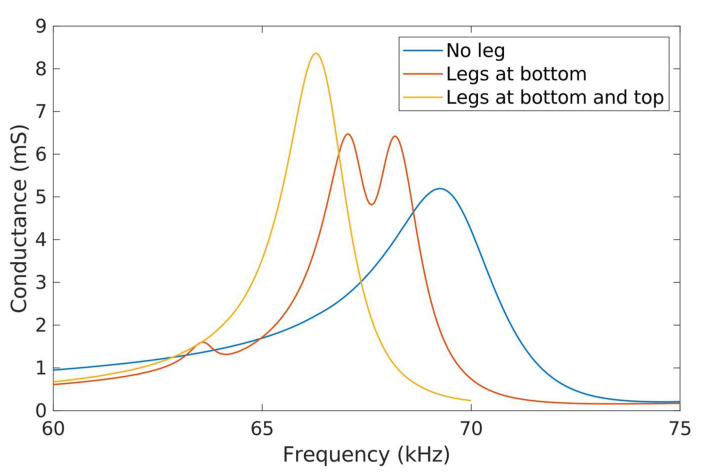
Measured electrical conductance of a PZT plate with no legs (blue), legs at the bottom of the plate (orange) and both the bottom and the top of the plate (yellow).

**Figure 10 micromachines-13-01815-f010:**
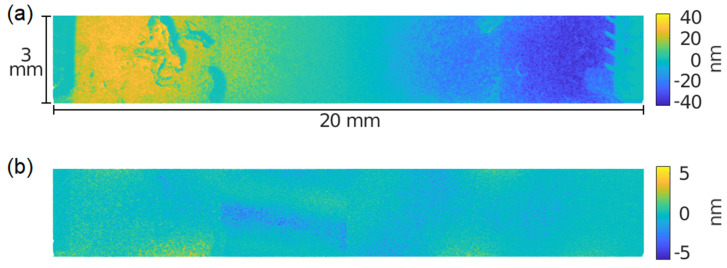
Optical measurement of the displacement of the first extensional mode of the PZT plate with legs at both the bottom and the top. (**a**) In-plane displacement. (**b**) Out-of-plane displacement.

**Figure 11 micromachines-13-01815-f011:**
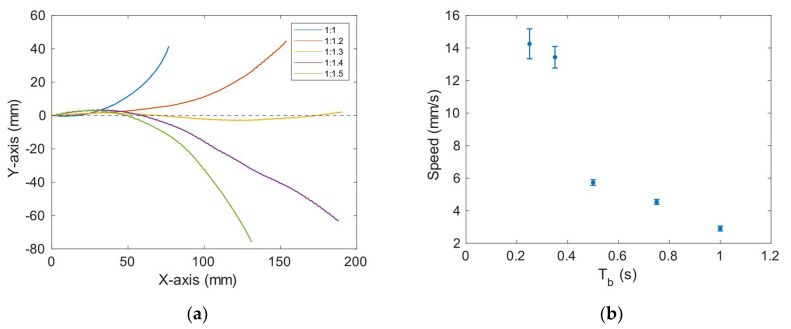
(**a**) Path followed by the robot depending on the T_ON_ ratio between plates. (**b**) Linear speed as a function of T_b_. Symbols represent the average of 3 measurements.

**Figure 12 micromachines-13-01815-f012:**
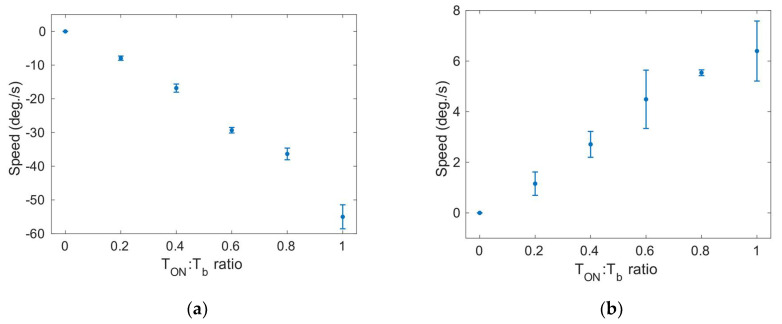
(**a**) Clockwise rotation speed as a function of T_ON_:T_b_ ratio. (**b**) Counterclockwise rotation speed as a function of T_ON_:T_b_ ratio. Symbols represent the average of 3 measurements.

**Figure 13 micromachines-13-01815-f013:**
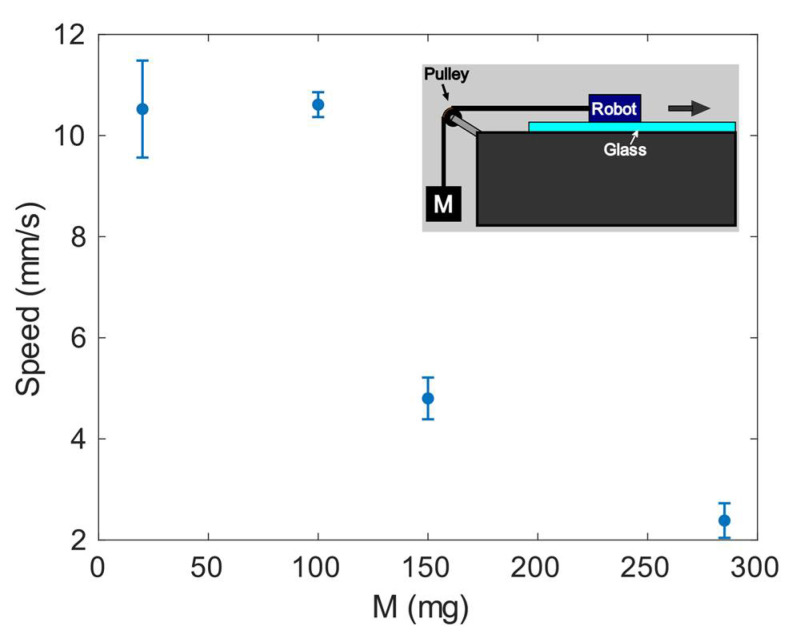
Speed versus pulled mass. Symbols represent the average of 3 measurements. Setup shown in the inset.

**Figure 14 micromachines-13-01815-f014:**
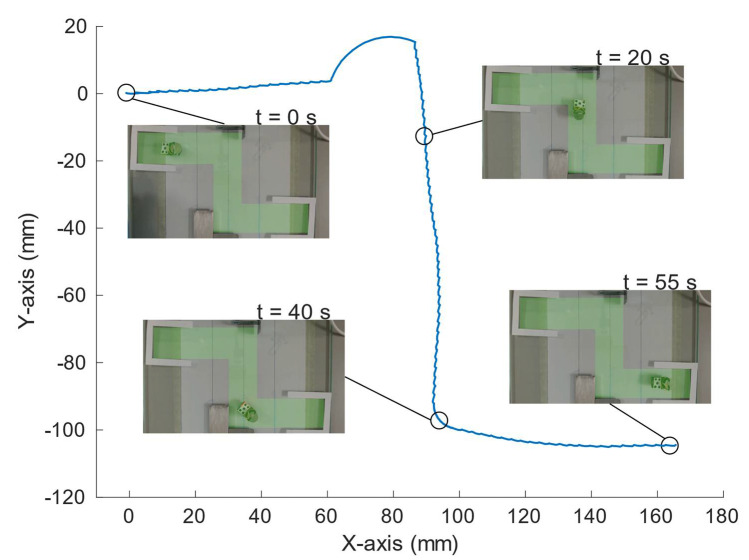
The path followed by the robot with photographs in different moments of the path.

**Table 1 micromachines-13-01815-t001:** Mass distribution of the robot.

Component	Mass (mg)
PZT plates	100 × 2 = 200
Microcontroller	1.123
PCB	874
Battery	3.270
Supporting platform	1.601
**Total Mass**	**7.068**

**Table 2 micromachines-13-01815-t002:** Calculated peak conductance (ΔG) and quality factor (Q) of the PZT plate for different leg combinations.

Conductance peak	ΔG (mS)	Q
No legs	17.28	69.15
Bottom legs (Peak 1)	9.18	89.54
Bottom legs (Peak 2)	10.82	87.59
Top and bottom legs	19.54	89.39

## Data Availability

Not applicable.

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
