# Peer review of "Miniature Autonomous Robot Based on Legged In-Plane Piezoelectric Resonators with Onboard Power and Control"

_micromachines, 2022, doi:10.3390/mi13111815_

Round 1

Reviewer 1 Report

Authors demonstrated a piezoelectric robot based on the first extensional resonant mode.

Authors emphasized that the robot vibrates on the first extensional mode. But the real excitation frequency of voltage signals seems different from the first extensional mode. Then, why not excite it by using the first extensional frequency in order to reach a faster speed?

From the aspect of my view, Eq. (1) is the extensional frequency of a free-free plate. However, the piezoelectric robot is not a free-free structure. For a more accurate simulation, the piezoelectric effect has to be considered for the calculation of the first extensional frequency. Similarly, for the displacement estimation in Eq. (2), a more accurate equation has to be considered because of piezoelectricity. With piezoelectricity, the first extensional frequency will be different. Will it influence the working efficiency of the present robot?

The finite element simulation presents two vibration modal shapes of different leg locations. Why two legs are put on the top of the plate? How do they influence the movement of the plate?

Please clarify the above issues

Author Response

Reply attached.

Reviewer 2 Report

as attached.

Author Response

Reply attached.

Round 2

Reviewer 1 Report

The authors addressed the most of my questions. The manuscript can be recommended for publication in this journal.